# Low genetic heterogeneity of *Leishmania major* in different geographical regions of Iran

**Adel Spotin** [ID][1,2]*, **Soheila Rouhani**[3], **Ali Haghighi**[3], **Parviz Parvizi**[4]

**1** Immunology Research Center, Tabriz University of Medical Sciences, Tabriz, Iran, **2** Department of Parasitology and Mycology, School of Medicine, Tabriz University of Medical Sciences, Tabriz, Iran, **3** Department of Medical Parasitology and Mycology, School of Medicine, Shahid Beheshti University of Medical Sciences, Tehran, Iran, **4** Molecular Systematics Laboratory, Parasitology Department, Pasteur Institute of Iran, Tehran, Iran

* Adelespotin@gmail.com, Spotina@tbzmed.ac.ir

**Data Availability Statement:** All relevant data are within the paper and its Supporting Information files.

## Abstract

To examine the genetic diversity of *Leishmania major*, 100 Giemsa-stained positive slides were collected from endemic foci of Iran (Northeast, Central, and Southwest provinces) over two consecutive years during 2019–2021. The *Leishmania* ITS-rDNA gene was amplified and *Leishmania* sp. was recognized by PCR-RFLP and sequencing. In addition, 178 registered ITS-rDNA sequences from other geographical regions of Iran were retrieved from GenBank, including different host species (human, sandfly and rodent). A total of 40 new haplotypes were discovered using the ITS-rDNA sequence analysis. IR29 (20.6%) and IR34 (61%) were the two most common haplotypes, represented by a star-like feature in the overall population. Analysis of the molecular variance test revealed low genetic diversity of *L. major* in human cases (Haplotype diversity; 0.341), rodent (Hd; 0.387) and sandfly (Hd; 0.390) sequences. The lowest genetic diversity of *L. major* was observed in Southwest/Southeast Iran (Hd: 0.104–0.286). The statistically *F*st value indicated that *L. major* is not genetically differentiated between geographic regions of Iran, except for the Northeast-Southwest (*F*st: 0.29055) and Central-Southwest (*F*st: 0.30294) population pairs. The current study as the first investigation discloses new perspectives for further evaluation in the identification local transmission paradigms and initiating effective prevention strategies.

## 1. Introduction

Cutaneous leishmaniasis (CL) is one of the neglected tropical diseases that is mainly spread in Iran by the causative agents of *Leishmania major* and *Leishmania tropica* [1, 2]. CL is prevalent in more than half of the Iranian provinces with a prevalence of 1.8% to 37.9% and an annual incidence of 26,630 cases/year [3, 4].

The spectrum of genetic (haplotype) variability in *L. major* is presented as one of the most controversial issues [1, 2, 5, 6]. The genetic heterogeneity of *Leishmania* species may result in the emergence of new species/strains/haplotypes and ultimately the creation of drug-resistant mutants [5, 6]. With regard to treatment failure, the identification of heterogeneity traits of *Leishmania* spp. and the emergence of resistant drug mutations should be addressed by policymakers.

**Funding:** This study was financially supported by Iran National Science Foundation: INSF, 99014249, Dr. Adel Spotin.

**Competing interests:** The authors have declared that no competing interests exist.

Several mitochondrial and nuclear DNA markers, including microsatellites, ribosomal internal transcribed spacer regions (ITS-rDNA), cytochrome c, 18S-rRNA, HSP-70, the gp63 gene locus, and minicircles of kinetoplast DNA were used to assess the genetic diversity of the *Leishmania* parasite [1, 7–12]. The ITS-rDNA is a useful and informative phylogenetic marker as it has sequences with different rates of evolutionary variability [5, 13]. A number of molecular assays have been employed to identify genetic polymorphism between inter-and intra-*Leishmania* species [14, 15]. These include multilocus microsatellite typing (MLMT), random amplification of polymorphic DNA (RAPD), restriction fragment length polymorphism (RFLP), multilocus enzyme electrophoresis (MLEE), multilocus sequence typing (MLST) and sequencing.

Several Iranian researchers have performed their molecular experiments on *L. major* genotyping in different host species (human, sandfly and rodent) and their results have shown that the new haplotypes circulate sympatrically in endemic regions of the country [16–19]. However, according to our literature review, there is no comparative study on the genetic diversity of *L. major* among different endemic foci of Iran.

Knowledge of the genetic characteristics of *L. major* in adjacent regions may provide the basis for conducting future epidemiological studies to implement control strategies. The aim of this study was to assess the genetic diversity and gene migration of *L. major* using ITS-rDNA nucleotide sequences derived from human, sandfly and rodent in endemic regions of Iran.

## 2. Materials and methods

### 2.1. Ethics statement and sample collection

All research involving human participants have been obtained in conformity with informed consent, privacy and confidentiality of patients who were sampled and analyzed anonymously during study. The majorities of the people were in the edge of Iran border and therefore could not able to get written consent one by one. Instead, the oral consent was obtained with the help of central public health authority as both mutual acquaintance and interpreter of the indigenous peoples. All experiments on the humans were performed according to the guidelines of the Ethical Board of Pasteur Institute of Iran.

One hundred positive microscopic slides from wet/dry lesions of confirmed CL patients were collected from endemic geographical regions of Iran (Southwest, Khuzestan Province; n: 40, Northeast, Golestan Province; n: 30 and Central; Isfahan Province; n: 30) from August 2019 to February 2021. In this study, the collected samples from indigenous cases were included and the imported cases from neighboring countries were excluded.

The positive slides were graded to rank leishman bodies' density from +1 to +6. In addition, in order to compare the analyzed sequences of the present study with five regions of Iran, 178 sequences of ITS-rDNA *L. major* including Southwest (Khuzestan, Ilam and Fars provinces), Central (Isfahan and Kashan provinces), North (Tehran Province), Southeast (Kerman), North-Central (Semnan), West (Lorestan and Kermanshah provinces) and Northeast (Golestan, Turkmen Sahra), Khorasan (Birjand, Esfarayen, Mashhad, Jajarm and Garmeh) populations were retrieved from the GenBank database for FASTA during 2008–2020.

### 2.2. DNA extraction, PCR amplification and PCR-RFLP for ITS-rDNA

DNA extraction of parasites was performed from Giemsa-stained positive slides (DNG Plus Kit, Iran) [20]. Single-round PCR was used to detect *Leishmania* sp. by amplifying ITS-rDNA about 480 bp. The details of PCR protocol and employed primers (ITS1F and ITS2R4) were the same as previously reported [1, 9, 17]. To digest the PCR amplicons in RFLP, the

endonuclease reaction of ITS-rDNA was performed in a volume of 30μL containing, 2μL of 10x buffer, 2μL of *BsuRI* (*HaeIII)* (cut site GG↓CC), 10μL of amplicon and 16μL of distilled water for 4 h at 37°C. The digested fragments were analyzed using electrophoresis on 1.5% agarose gel containing safe stain and ladder of 100 bp.

## 2.3. DNA sequencing, data extraction and bioinformatic analysis of ITS-rDNA

Forty PCR products were successfully sequenced using the ITS primers, ITS1F and ITS2R4 (Codon Company, Iran). Sequences were trimmed and edited in consensus positions compared to regional sequences using Sequencher v.5.4.6 software. Furthermore, 178 registered ITS-rDNA nucleotide sequences of *L. major* from other geographic populations of Iran including different host species (human, sandfly and rodent) were downloaded from GenBank. The analysis of molecular variance (AMOVA) was performed by DnaSP software to determine the genetic diversity indices (number of haplotypes (Hn); nucleotide diversity (Nd); haplotype diversity (Hd), neutrality indices (Tajima's *D* and Fu's Fs tests) [21]. The pairwise fixation index (*F*st: F-statistics) and number of migration (Nm) were obtained to assess the gene migration of *L. major* between different geographical regions of Iran. To ascertain the genealogical relationships of intra-species diversity of *L. major*, a haplotype network was drawn by PopART software determined by the Median-Joining model [22]. To confirm taxonomic status of the *L. major*, a phylogenetic tree was constructed using the program Splits Tree v.4.0 based on the Neighbor Net method and Median-Joining analysis. Distance scale was 0.01 indicating the number of base substitutions per site. Also *Trypanosoma brucei* (Accession number; **JN673390**) was considered as an outgroup branch.

## 3. Results

### 3.1. Identification of Leishmania sp. using PCR-RFLP

The clinical presentations of wet/dry lesions were classified into classic (volcanic form) and non-classic (psoriasiform, herpetiform, hyperkeratotic, eczematoid, zosteriform and erythematous papulonodules signs). The ITS-rDNA gene (480 bp) was amplified from all 100-CL stained slides. Based on the PCR-RFLP assay, all *Leishmania* PCR products were digested by *BsuRI (HaeIII)* and then two fragments of 140 bp and 340 bp were definitely assigned to *L. major*. To confirm the PCR-RFLP findings, 40 *L. major* amplicons were successfully sequenced, trimmed and edited to construct the phylogenetic tree.

### 3.2. Determination of genetic diversity and genetic differentiation between L. major

The edited sequences (n: 40) were compared with sequences obtained (n: 178) from the GenBank database. Ten sequences (including new haplotypes) were submitted to the GenBank database, and accession numbers were presented in S1 Table. A total of 40 new haplotypes were identified based on sequence analysis of the ITS-rDNA gene (S1 Table). According to AMOVA test, low genetic heterogeneity of *L. major* was found in Southwest (Hd: 0.104, Hn: 5), Southeast (Hd: 0.286, Hn: 2), Central (Hd: 0.419, Hn: 10), in the west (Hd: 0.334, Hn: 4), Northeast (Hd: 0.429, Hn: 17) and North-Central Iran (Hd: 0.400, Hn: 2) (Table 1).

Based on host species, low genetic (haplotype) diversity of *L. major* was observed in reservoir rodents (*Rhombomys opimus* and *Meriones libycus*, Hd; 0.387, Hn: 10) and sandflies (*Phlebotomus papatasi/Phlebotomus caucasicus*, Hd; 0.390, Hn: 7) compared to human CL cases (Hd; 0.441, Hn: 23) (Table 2). The highest nucleotide diversity was found in Central Iran (Nd:

**Table 1.  Diver sity and neutrality indices of *L. major* using ITS-rDNA sequences in various geographical regions of Iran.**

| Region | Province/cities (n) | Diversity indices | | | | | | Neutrality indices | |
|---|---|---|---|---|---|---|---|---|---|
| | | N | Hn | Hd± SD | Polymorphic sites | Nd (π) | K | Tajima's D* | Fu's Fs statistic** |
| **Central Iran** | Isfahan (16), Kashan (14) (Aran Bidgol) | 30 | 10 | 0.419 ±0.080 | 17 | 0.01005 | 2.724 | -1.49695 | -9.012 |
| **Southwest Iran** | Fars (1), Ilam (37), Khuzestan(10) | 48 | 5 | 0.104 ±0.067 | 31 | 0.00894 | 2.083 | -2.56931 | 4.546 |
| **Northeast Iran** | Khorasan (Birjand; 25, Esfarayen; 31, Mashhad; 13, Jajarm; 5, Garmeh; 3), Golestan (37; Turkmen sahra) | 114 | 17 | 0.429± 0.059 | 41 | 0.00901 | 2.459 | -2.27541 | -9.662 |
| **Southeast Iran** | Kerman (n; 7, Bam, Jiroft and Shahdad) | 7 | 2 | 0.286± 0.196 | 4 | 0.00351 | 1.143 | -1.43414 | 2.047 |
| **West Iran** | Lorestan (8), Kermanshah (6; Ghasre shirin) | 14 | 4 | 0.334 ±0.045 | 29 | 0.02279 | 6.154 | -1.75475 | -0.796 |
| **North central Iran** | Semnan (Damghan; 5) | 5 | 2 | 0.400± 0.175 | 1 | 0.00202 | 0.600 | 1.22474 | 0.626 |
| **Total** | | 218 | 40 | | | | | | |

N: number of isolates; Hn: number of haplotypes; Hd: haplotype diversity; Nd: nucleotide diversity; K: Average number of nucleotide differences.*Significant, $P < 0.01$**Significant, $P < 0.02$

0.01005) particularly in reservoir rodents (Nd: 0.01164) (Tables 1 and 2). Neutrality indices of ITS-rDNA showed negative values (-2.27541 for Tajima's D to -9.662 for Fu's Fs statistic) in Northeast, West and Central populations which indicating a deviation from neutrality (Table 1). Within 480 bp consensus position of ITS-rDNA, 48 point mutations were identified. 24 of these were parsimony-informative sites and 24 of these were singleton variable sites. The statistically *F*st value showed that *L. major* is not genetically differentiated between geographic regions of Iran (*F*st: 0.01649 to 0.07820) except Northeast-Southwest (*F*st: 0.29055, Nm: 0.61) and Central-Southwest (*F*st: 0.30294, Nm: 0.58) population pairs (Table 3).

## 3.3. Phylogenetic tree and haplotype network of ITS-rDNA

Neighbor Net tree generated by ITS-rDNA sequences demonstrated that *L. major* identified

**Table 2.  Diversity and neutrality indices of *L. major* in different host species (human, sandfly and rodent) using ITS-rDNA sequences in various geographical regions of Iran.**

| Parasite | Host-vector | | Diversity indices | | | | | | Neutrality indices | |
|---|---|---|---|---|---|---|---|---|---|---|
| | | | N | Hn | Hd± SD | Nd (π) | Polymorphic site | K | Tajima's D* | Fu's Fs statistic** |
| ***Leishmania major*** | **Vector (Sandfly)** | ***Phlebotomus papatasi/ Phlebotomus caucasicus*** | 25 | 7 | 0.390±0.079 | 0.00565 | 8 | 1.600 | -0.77730 | -1.466 |
| | **Reservoir (Vertebrate hosts)** | Human | 168 | 23 | 0.441 ± 0.030 | 0.00426 | 23 | 1.205 | -2.03615 | -29.979 |
| | | Rodents: ***Rhombomys opimus*** and ***Meriones libycus*** | 25 | 10 | 0.387± 0.105 | 0.01164 | 29 | 3.923 | -1.95708 | -2.662 |
| **Total** | | | 218 | 40 | | | | | | |

* Significant, $P<0.01$

**Significant, $P <0.02$.

N: number of isolates; Hn: number of haplotypes; Hd: haplotype diversity; Nd: nucleotide diversity; K: Average number of nucleotide differences.

**Table 3. Pairwise fixation index (*F*st) for *L. major* isolates originating from five regions of Iran using nucleotide sequence data of ITS-rDNA.**

| Region | | Populations | | | |
|---|---|---|---|---|---|
| | Province (cities) | Central Iran | Southwest Iran | West Iran | Northeast Iran |
| **Central Iran** | Isfahan and Kashan (Aran Bidgol) | | Nm: 0.58 | Nm: 6.52 | Nm: 14.91 |
| **Southwest Iran** | Fars, Ilam and Khuzestan | 0.30294 | - | Nm: 2.95 | Nm: 0.61 |
| **West Iran** | Lorestan, Kermanshah (Ghasre shirin) | 0.03692 | 0.07820 | - | Nm: 2.50 |
| **Northeast Iran** | Khorasan (Birjand, Esfarayen, Mashhad, Jajarm and Garmeh), Golestan (Turkmen sahra) | 0.01649 | 0.29055 | 0.09078 | - |

(Accession nos; **OP811334, OP811489, OP811525, OP829807-OP829812**) assigned into *L. major* complex (Fig 1). The identified haplotypes (IR1-1R40) based on their different host species are presented in Fig 2. A total of 7, 10 and 23 haplotypes of *L. major* were detected in the sandfly, rodent and human, respectively in the different CL foci of Iran (Fig 2 and Table 2). IR29 (20.6%) and IR34 (61%) were the two most common haplotypes in the whole population, represented by a star-like feature in the overall population.

Occurrence of haplotypes of IR28 (Accession no: **KF899848**) and IR31 (Accession no: **KP773410**) between common haplotypes of IR29 and IR34 suggests that human-derived *L. major* may circulate between geographic regions of Iran (Fig 2).

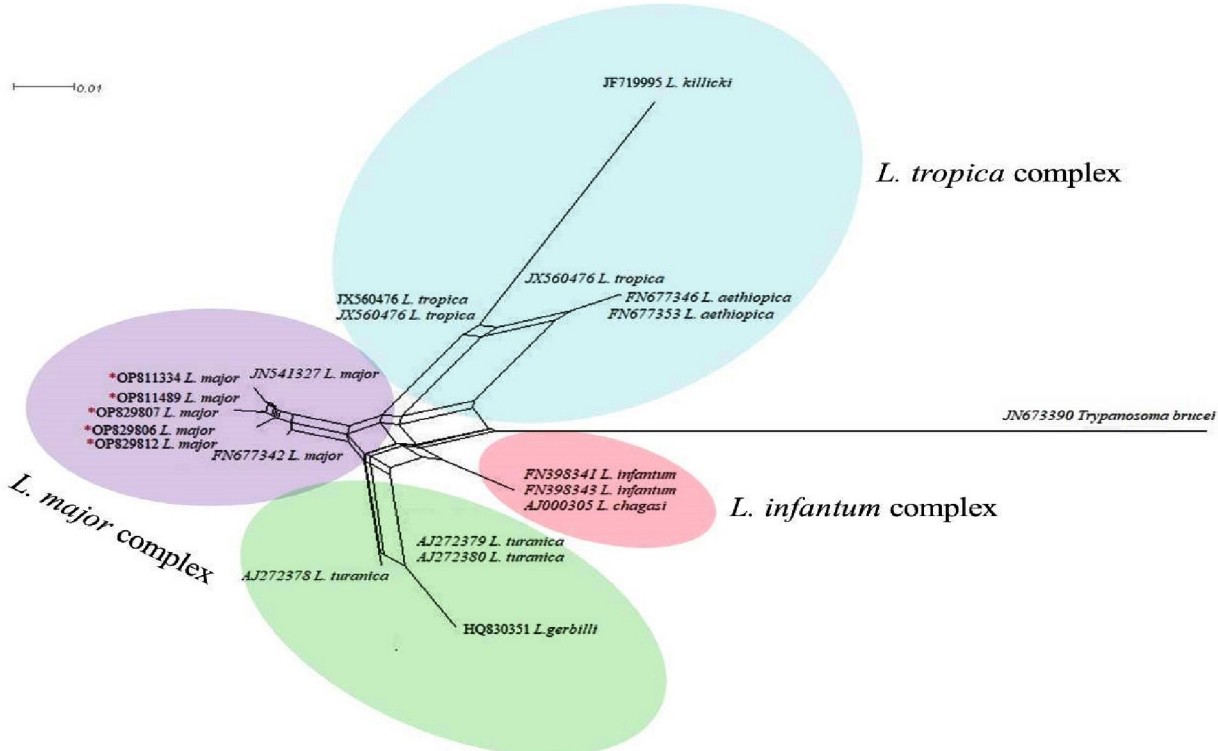

**Fig 1. Neighbor-Net graph drawn by *Leishmania* spp. by using the Splits Tree 4.0 program.** The identified isolates of *L. major* in this study marked by an asterisk (*). *Trypanosoma brucei* (Accession no: **JN673390**) was considered as out-group branch. The distance scale was estimated 0.01.

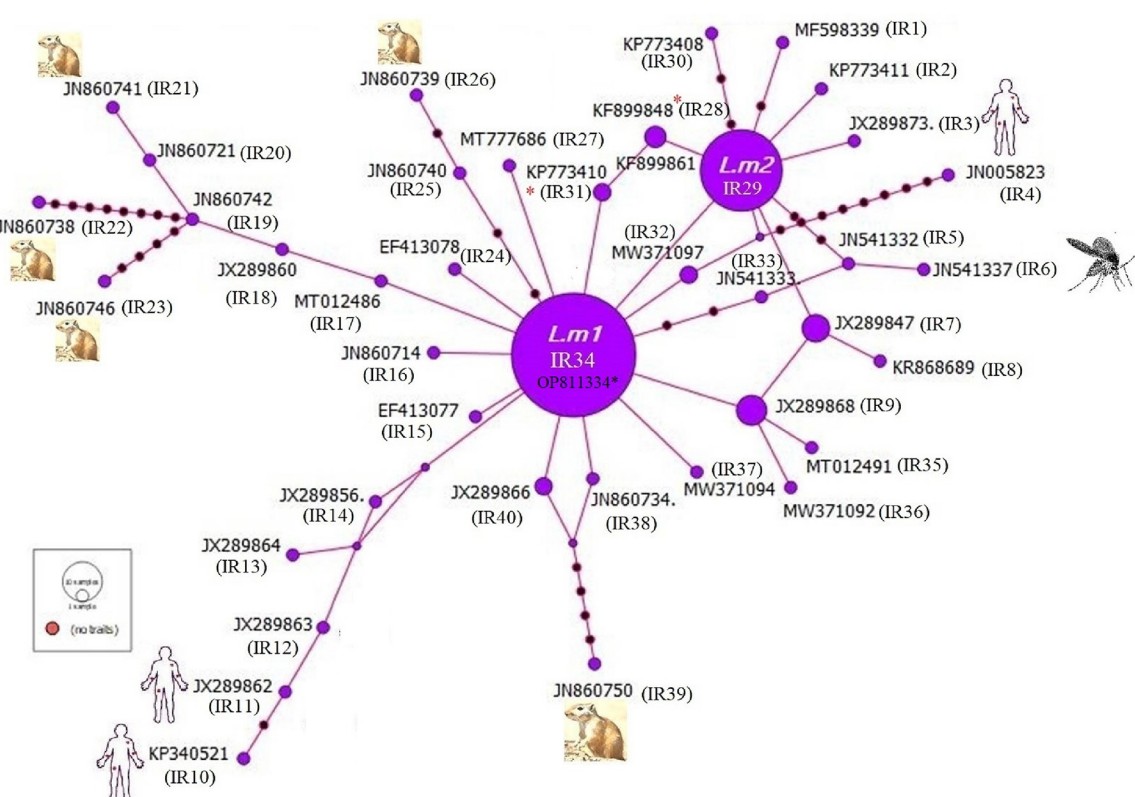

**Fig 2. ITS-rDNA haplotype network in *L. major* from different host species (human, sandfly and rodent) in various geographical foci of Iran.** Submitted sequences (n: 10) in this study were considered as common haplotype L. m1 (**OP811334***). The red circles are relative to the frequency of each haplotype. Each line between haplotypes indicate single mutational step.

## 4. Discussion

Knowledge of *L. major* genetic diversity of is essential to design epidemiological studies to implement monitoring, surveillance and control programs [5, 6]. This study analyzed the nucleotide sequence data on the heterogeneous population structure of *L. major* originating from sandfly, rodent and human hosts in Iran where different ecological conditions occur in widely separated regions.

In this study, the ITS-rDNA marker was used to detect the evolutionary relationship of *L. major*, as this DNA barcode marker can depict a reliable picture of the various phylogenetic subdivisions within this genus [13].

Based on current findings, micro-heterogeneity of *L. major* observed in reservoir rodents (*R. opimus* and *M. libycus*), sandfly vectors (*Ph. papatasi/Ph. caucasicus*) and human CL cases. This outcome indicates that rodents, sandflies, and human hosts are unlikely to exert a selective pressure on *L. major* and emphasizes the geographic/ecologic dependence of the low genetic heterogeneity in Iran. On the other hand, low genetic diversity of *L. major* in human, rodent and sandfly sequences can be explained by the genomic features of ITS-rDNA. In fact, the GC content of *L. major* (59.7%) is generally higher than that of *L. tropica*, so the stability of the triple hydrogen bonds of the GC pair and the stacking interaction are subjected to slippage [23, 24].

In a related study, low genetic heterogeneity of *L. major* isolates collected from patients and rodents was identified by MLMT technique in emergent foci of zoonotic cutaneous leishmaniasis in Tunisia [25, 26].

However, Tashakori et al., (2011) have shown three distinct genetic clusters of *L. major* (n: 26 isolates) using MLMT in Iran and they stated that significant genetic diversity of *L. major* is related to the existence of different populations of *Ph. papatasi* and/or differences in the abundance of reservoir hosts in different regions of Iran [27].

Some evidence suggests that the level of genetic diversity in the *Leishmania* parasite depends on the effective population size and is higher in small populations [5]. Furthermore, environmental alterations and biotic interactions among five regions of Iran can exert strong selective pressures on different life-history aspects and hence influence their degree of genetic diversity in *Leishmania* parasite [5]. However, other authors have reported a decreasing degrees of genetic diversity among the following *Leishmania* spp.; *L. tropica*> *L. aethiopica*> **L. major**> *L. donovani* [28].

In this study, a variety of clinical manifestations (classical and non-classical forms) of *L. major* were observed in human lesions, however no significant genetic diversity (Hd; 0.541) was found among ITS-rDNA sequences of *L. major* in Khuzestan, Southwest Iran. These findings attenuate relationship between phenotypic and genotypic traits of CL in *L. major* patients, possibly demonstrating that genetic heterogeneity may not definitely influence the formation of various clinical manifestations.

The negative neutrality indices for the *L. major* population indicate evidence of some likely mechanisms including the neutral mutation model, population size equilibrium and purifying selection [29].

The significant $F$st values (0.29055 to 0.30294) of ITS-rDNA sequences showed that *L. major* Northeast-Southwest and Central-Southwest populations were genetically differentiated. It could be postulated that no gene migration of *L. major* taken place between local isolates of the mentioned regions. This also indicates that the diversity of the *L. major* population was unlikely to be affected by the bottleneck effect.

Furthermore, the occurrence of distinct genetic structure of *L. major* between mentioned regions indicates that this protozoan may have been sustained by indigenous human, rodents and sandflies during the era. On the other hand, low values of $F$st (0.01649 to 0.07820) showed that *L. major* West-Central, Central-Northeast, West-Southwest populations were not genetically differentiated, indicating a gene migration of *L. major* has probably occurred between mentioned population pairs.

The emergence of haplotypes of IR28 (human isolate) and IR31 (human isolate) between common haplotypes of IR29 and IR34 shows that here is dawn of *L. major* migration as a result of transmission of haplotypes from one population to another population through ecological changes and/or host mobility.

Charyyeva et al., (2021) demonstrated that *L. tropica* isolated from Syrian and Turkish patients using ITS1 sequences exhibits a complex phylo-geographic pattern, with some haplotypes being widespread across the Turkey and Syria [30].

One of the limitations of the current study was that the *L. major* sequences obtained from sandflies and rodents were partially small, in order to infer the extensive genetic diversity on a large scale.

In conclusion, the present study as the first investigation strengthens our knowledge of local transmission paradigms and the genetic data of *L. major* in different geographical regions of Iran. Moreover low genetic heterogeneity of *L. major* in human, rodent and sandfly hosts should be highlighted as a treatment target for the emergence of probable drug-resistant mutants, particularly in clinical *Leishmania* strains. Further research is needed to develop next-generation sequencing of *Leishmania* sp. through the use of informative markers in larger areas of Iran and surrounding countries to assess the potential evolutionary scenario.

## Supporting information

**S1 Table. Identified haplotypes of *L. major* based on ITS-rDNA sequences in various geographical foci of Iran.**
(DOCX)

## Acknowledgments

The collections of microscopic slides were made possible by the assistance of the Centre of Health Services Khuzeastan and Golestan. The authors thank Mehdi Baghban for helping with the field work and Elnaz AlaeeNovin for helping in Molecular Systematics Laboratory.

## Author Contributions

**Conceptualization:** Adel Spotin, Parviz Parvizi.

**Formal analysis:** Adel Spotin, Ali Haghighi.

**Investigation:** Soheila Rouhani, Parviz Parvizi.

**Methodology:** Adel Spotin, Parviz Parvizi.

**Supervision:** Adel Spotin, Soheila Rouhani, Parviz Parvizi.

**Writing – original draft:** Adel Spotin.

**Writing – review & editing:** Soheila Rouhani, Ali Haghighi, Parviz Parvizi.

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
