## [Decision Letter · Decision Letter 0]

27 Feb 2023

PONE-D-23-00609Low genetic heterogeneity of Leishmania major in various geographical regions of IranPLOS ONE

Dear Dr. Spotin,

Thank you for submitting your manuscript to PLOS ONE. After careful consideration, we feel that it has merit but does not fully meet PLOS ONE’s publication criteria as it currently stands. Therefore, we invite you to submit a revised version of the manuscript that addresses the points raised during the review process.

We look forward to receiving your revised manuscript.

Kind regards,

Alireza Badirzadeh

Academic Editor

PLOS ONE

Journal Requirements:

6. We note that Figure 1 in your submission contain [map/satellite] images which may be copyrighted. All PLOS content is published under the Creative Commons Attribution License (CC BY 4.0), which means that the manuscript, images, and Supporting Information files will be freely available online, and any third party is permitted to access, download, copy, distribute, and use these materials in any way, even commercially, with proper attribution. For these reasons, we cannot publish previously copyrighted maps or satellite images created using proprietary data, such as Google software (Google Maps, Street View, and Earth). For more information, see our copyright guidelines: http://journals.plos.org/plosone/s/licenses-and-copyright.

Reviewers' comments:

Reviewer's Responses to Questions

**Comments to the Author**

1. Is the manuscript technically sound, and do the data support the conclusions?

Reviewer #1: Yes

Reviewer #2: Yes

Reviewer #3: Yes

2. Has the statistical analysis been performed appropriately and rigorously? 

Reviewer #1: Yes

Reviewer #2: I Don't Know

Reviewer #3: Yes

3. Have the authors made all data underlying the findings in their manuscript fully available?

Reviewer #1: No

Reviewer #2: Yes

Reviewer #3: Yes

4. Is the manuscript presented in an intelligible fashion and written in standard English?

Reviewer #1: No

Reviewer #2: Yes

Reviewer #3: Yes

5. Review Comments to the Author

Reviewer #1: Authors investigated the genetic diversity of Leishmania major, 100 positive stained slides were collected from endemic foci of Iran (Northeastern, Central, and Southwestern provinces) over two consecutive years during 2019-2021. In general this is well-written manuscript, important for the field and for the region. I suggest to cut Introduction and to focus only on the topic, also please be focused in Discussion. Methodology is well written. English should be revised by English speaker.

Reviewer #2: The manuscript presents a good concept; however, some points are not clear to the reviewer which are summarized below:

In Abstract

-In line 15: “100 positive stained slides” please determine, Giemsa or Wright stained slides?

-In line 24, please correct the spelling of "Haplotype". And check the spelling of all of the text.

-Please “PCR-RFLP” added to the Keywords.

In introduction

-Considering the low genetic diversity of L. major was observed in other studies in Iran, please explain the novelty and necessity of conducting this study.

-In lines 49-53: Despite the existence of a number of molecular assays for identifying genetic polymorphism, why did you choose the "RFLP" method among them?

Despite the existence of a number of methods, why did you use the "L" method among them?

In Materials and methods

-According to the wideness of the selected geographical areas, the number of 100 slides is not suitable. Was there a reason for not providing more slides?

-What inclusion and exclusion criteria are considered in the selection of patients? For example, are immigrants or imported cases considered?

-In lines 65 and 86: Please write the number in the form of a letter at the beginning of the sentence, for example, "one hundred".

In discussion

-In line 148: the full form of ITS-rDNA has already been mentioned, please just write the abbreviation form.

-Please mention the limitations of this research in the discussion section.

Reviewer #3: The manuscript entitled "Low genetic heterogeneity of Leishmania major in various geographical regions of Iran" describes the diversity of L. major isolates and its implication on the epidemiology of zoonotic cutaneous leishmaniasis in Iran. However, several points must be addressed. Thus, I recommend that this manuscript should be published only after considering these major following changes.

Specific Comments

- All the Tables can go to supplements.

- Please explain why only 40 amplicons (out of 100) were sequenced?

- Please put your research in the context of similar studies in other Leishmania spp. from the same region.

- The authors should discuss further the results obtained by this method used in this study compared to the findings of other studies performed by using MLMT or NGS reporting contrasting results.

- The authors referred to one study instead to reporting the results of most studies performed on L. major.

- The authors should discuss the findings of other studies performed in Turkey and Syria (entitled: Genetic diversity of Leishmania tropica: Unexpectedly complex distribution pattern; F.e. doi:10.1016/j.actatropica.2021.105888.) by comparison with the results of this study.

- - Resolution of figures must be improved.

- Line 68: The location of Fig 2 don’t support in the text.

6. PLOS authors have the option to publish the peer review history of their article (what does this mean?). If published, this will include your full peer review and any attached files.

Reviewer #1: **Yes: **Aleksandra Barac

Reviewer #2: No

Reviewer #3: No

---

## [Author Response · Author response to Decision Letter 0]

27 Mar 2023

To: Plos One

Dear Dr. Alireza Badirzadeh

This is my FIRST revision of manuscript

Ms. Ref. No.: PONE-D-23-00609

Title: “Low genetic heterogeneity of Leishmania major in various geographical regions of Iran”

Reviewers' comments:

Reviewer's Responses to Questions

Comments to the Author

Reviewer #1: Authors investigated the genetic diversity of Leishmania major, 100 positive stained slides were collected from endemic foci of Iran (Northeastern, Central, and Southwestern provinces) over two consecutive years during 2019-2021. In general this is well-written manuscript, important for the field and for the region. I suggest to cut Introduction and to focus only on the topic, also please be focused in Discussion. Methodology is well written. English should be revised by English speaker.

Response: Thank you. Discussion was revised and English was polished entire MS.

Reviewer #2: The manuscript presents a good concept; however, some points are not clear to the reviewer which are summarized below:

In Abstract

-In line 15: “100 positive stained slides” please determine, Giemsa or Wright stained slides?

Response: The “100 Giemsa stained positive” was highlighted in line15.

-In line 24, please correct the spelling of "Haplotype". And check the spelling of all of the text.

Response: It was done and highlighted in line 24.

-Please “PCR-RFLP” added to the Keywords. Response: It was done and highlighted

In introduction

-Considering the low genetic diversity of L. major was observed in other studies in Iran, please explain the novelty and necessity of conducting this study.

Response: In this study, for the first time, haplotype distribution and gene migration of Leishmania major were assessed among various sources of human, rodent and sandfly in different geographical regions of Iran which can depict a real picture of epidemiology of parasite.

-In lines 49-53: Despite the existence of a number of molecular assays for identifying genetic polymorphism, why did you choose the "RFLP" method among them? Response: Among the known molecular methods, the PCR-RFLP for ITS-rDNA using BsuRI (HaeIII) enzyme has widely used because of the ease in differentiating Leishmania major, L. tropica and L. infantum.

In Materials and methods

-According to the wideness of the selected geographical areas, the number of 100 slides is not suitable. Was there a reason for not providing more slides?

Response: In this study, we were able to collect 100 positive slides during 2019-2021. To solve this gap, the sequences recorded (GenBank) from Iran was used to compare genomic analyzes.

-What inclusion and exclusion criteria are considered in the selection of patients? For example, are immigrants or imported cases considered?

Response: In this study, the collected samples from indigenous cases were included and the imported cases from neighboring countries were excluded (Lines 67-68).

-In lines 65 and 86: Please write the number in the form of a letter at the beginning of the sentence, for example, "one hundred". Response: Thank you. It was done and highlighted.

In discussion

-In line 148: the full form of ITS-rDNA has already been mentioned, please just write the abbreviation form. Response: It was done and highlighted.

-Please mention the limitations of this research in the discussion section. Response: It was mentioned in line 193.

Reviewer #3: The manuscript entitled "Low genetic heterogeneity of Leishmania major in various geographical regions of Iran" describes the diversity of L. major isolates and its implication on the epidemiology of zoonotic cutaneous leishmaniasis in Iran. However, several points must be addressed. Thus, I recommend that this manuscript should be published only after considering these major following changes.

Specific Comments

- All the Tables can go to supplements. Response: Due to the fact that tables 2 to 4 are directly related to the explanations of the results section of the article, therefore table 1 was considered as supplementary file.

- Please explain why only 40 amplicons (out of 100) were sequenced? Response: In this study, only 40 samples were successfully sequenced due to limited financial resources, bad sequencing and permanent non-specific bands.

- Please put your research in the context of similar studies in other Leishmania spp. from the same region. Response: It was done in conclusion and highlighted. 

- The authors should discuss further the results obtained by this method used in this study compared to the findings of other studies performed by using MLMT or NGS reporting contrasting results. Response: It was done (lines 161-163, 164-167).

- The authors referred to one study instead to reporting the results of most studies performed on L. major. Response: It was discussed and highlighted in discussion.

- The authors should discuss the findings of other studies performed in Turkey and Syria (entitled: Genetic diversity of Leishmania tropica: Unexpectedly complex distribution pattern; F.e. doi:10.1016/j.actatropica.2021.105888.) by comparison with the results of this study.

Response: It was discussed and highlighted in lines 194-196.

- - Resolution of figures must be improved. Response: The resolution of Figs was improved to 300 dpi.

- Line 68: The location of Fig 2 don’t support in the text. Response: Thank you. It was corrected.

  

Kind Regards

Adel Spotin, Ph.D.

*Corresponding author: Immunology Research Center, Tabriz University of Medical Sciences, Tabriz, Iran, Email: Adelespotin@gmail.com, Spotina@tbzmed.ac.ir

---

## [Decision Letter · Decision Letter 1]

26 Apr 2023

Low genetic heterogeneity of Leishmania major in different geographical regions of Iran

PONE-D-23-00609R1

Dear Dr. Adel Spotin,

We’re pleased to inform you that your manuscript has been judged scientifically suitable for publication and will be formally accepted for publication once it meets all outstanding technical requirements.

Kind regards,

Alireza Badirzadeh

Academic Editor

PLOS ONE

Additional Editor Comments (optional):

Reviewers' comments:

Reviewer's Responses to Questions

**Comments to the Author**

1. If the authors have adequately addressed your comments raised in a previous round of review and you feel that this manuscript is now acceptable for publication, you may indicate that here to bypass the “Comments to the Author” section, enter your conflict of interest statement in the “Confidential to Editor” section, and submit your "Accept" recommendation.

Reviewer #2: All comments have been addressed

Reviewer #3: All comments have been addressed

2. Is the manuscript technically sound, and do the data support the conclusions?

Reviewer #2: Yes

Reviewer #3: Yes

3. Has the statistical analysis been performed appropriately and rigorously? 

Reviewer #2: Yes

Reviewer #3: Yes

4. Have the authors made all data underlying the findings in their manuscript fully available?

Reviewer #2: Yes

Reviewer #3: Yes

5. Is the manuscript presented in an intelligible fashion and written in standard English?

Reviewer #2: Yes

Reviewer #3: Yes

6. Review Comments to the Author

Reviewer #2: (No Response)

Reviewer #3: The authors carried out all corrections in the manuscript. It seems the manuscript is appropriate for publication in this journal.

7. PLOS authors have the option to publish the peer review history of their article (what does this mean?). If published, this will include your full peer review and any attached files.

Reviewer #2: No

Reviewer #3: No

---

## [Editor Report · Acceptance letter]

28 Apr 2023

PONE-D-23-00609R1 

Low genetic heterogeneity of *Leishmania major* in different geographical regions of Iran 

Dear Dr. Spotin:

I'm pleased to inform you that your manuscript has been deemed suitable for publication in PLOS ONE. Congratulations! Your manuscript is now with our production department. 

Kind regards, 

on behalf of

Dr. Alireza Badirzadeh 

Academic Editor

PLOS ONE